# Do Mass Spectrometry-Derived Metabolomics Improve the Prediction of Pregnancy-Related Disorders? Findings from a UK Birth Cohort with Independent Validation

**DOI:** 10.3390/metabo11080530

**Published:** 2021-08-10

**Authors:** Nancy McBride, Paul Yousefi, Ulla Sovio, Kurt Taylor, Yassaman Vafai, Tiffany Yang, Bo Hou, Matthew Suderman, Caroline Relton, Gordon C. S. Smith, Deborah A. Lawlor

**Affiliations:** 1MRC Integrative Epidemiology Unit, University of Bristol, Bristol BS8 2BN, UK; paul.yousefi@bristol.ac.uk (P.Y.); kurt.taylor@bristol.ac.uk (K.T.); matthew.suderman@bristol.ac.uk (M.S.); caroline.relton@bristol.ac.uk (C.R.); d.a.lawlor@bristol.ac.uk (D.A.L.); 2NIHR Bristol Biomedical Research Centre, University of Bristol, Bristol BS8 2BN, UK; 3Department of Population Health Sciences, University of Bristol, Bristol BS8 2BN, UK; 4NIHR Cambridge Biomedical Research Centre, Department of Obstetrics and Gynaecology, University of Cambridge, Cambridge CB2 0QQ, UK; us253@medschl.cam.ac.uk (U.S.); gcss2@cam.ac.uk (G.C.S.S.); 5Bradford Institute for Health Research, Bradford Teaching Hospitals NHS Foundation Trust, Bradford BD9 6DA, UK; yassaman.vafai@york.ac.uk (Y.V.); tiffany.yang@bthft.nhs.uk (T.Y.); bo.hou@bthft.nhs.uk (B.H.)

**Keywords:** prediction, pregnancy, metabolomics, metabolites, mass spectrometry

## Abstract

Many women who experience gestational diabetes (GDM), gestational hypertension (GHT), pre-eclampsia (PE), have a spontaneous preterm birth (sPTB) or have an offspring born small/large for gestational age (SGA/LGA) do not meet the criteria for high-risk pregnancies based upon certain maternal risk factors. Tools that better predict these outcomes are needed to tailor antenatal care to risk. Recent studies have suggested that metabolomics may improve the prediction of these pregnancy-related disorders. These have largely been based on targeted platforms or focused on a single pregnancy outcome. The aim of this study was to assess the predictive ability of an untargeted platform of over 700 metabolites to predict the above pregnancy-related disorders in two cohorts. We used data collected from women in the Born in Bradford study (BiB; two sub-samples, *n* = 2000 and *n* = 1000) and the Pregnancy Outcome Prediction study (POPs; *n* = 827) to train, test and validate prediction models for GDM, PE, GHT, SGA, LGA and sPTB. We compared the predictive performance of three models: (1) risk factors (maternal age, pregnancy smoking, BMI, ethnicity and parity) (2) mass spectrometry (MS)-derived metabolites (*n* = 718 quantified metabolites, collected at 26–28 weeks’ gestation) and (3) combined risk factors and metabolites. We used BiB for the training and testing of the models and POPs for independent validation. In both cohorts, discrimination for GDM, PE, LGA and SGA improved with the addition of metabolites to the risk factor model. The models’ area under the curve (AUC) were similar for both cohorts, with good discrimination for GDM (AUC (95% CI) BiB 0.76 (0.71, 0.81) and POPs 0.76 (0.72, 0.81)) and LGA (BiB 0.86 (0.80, 0.91) and POPs 0.76 (0.60, 0.92)). Discrimination was improved for the combined models (compared to the risk factors models) for PE and SGA, with modest discrimination in both studies (PE-BiB 0.68 (0.58, 0.78) and POPs 0.66 (0.60, 0.71); SGA-BiB 0.68 (0.63, 0.74) and POPs 0.64 (0.59, 0.69)). Prediction for sPTB was poor in BiB and POPs for all models. In BiB, calibration for the combined models was good for GDM, LGA and SGA. Retained predictors include 4-hydroxyglutamate for GDM, LGA and PE and glycerol for GDM and PE. MS-derived metabolomics combined with maternal risk factors improves the prediction of GDM, PE, LGA and SGA, with good discrimination for GDM and LGA. Validation across two very different cohorts supports further investigation on whether the metabolites reflect novel causal paths to GDM and LGA.

## 1. Introduction

Gestational diabetes (GDM), gestational hypertension (GHT), pre-eclampsia (PE), small for gestational age (SGA), large for gestational age (LGA) and spontaneous preterm birth (sPTB) are common pregnancy-related disorders [1,2,3,4,5,6,7]—associated with long-term mortality and morbidity in mother and offspring [7,8,9,10]. Currently, the prediction of these disorders relies largely on stratifying women based on established risk factors. The established risk factors for predicting these disorders include maternal smoking [11], age [12], ethnicity [13], parity [14] and body mass index (BMI) [15]. However, many women do not meet any of these risk factors and yet go on to have a complicated pregnancy [16,17,18]. A good indicator of risk is previous pregnancy history. However, this is not obtainable in nulliparous women [19,20], and there is a need for clinical prediction models that do not depend on previous pregnancy history [19]. Development of such might result in better ways of managing antenatal care by intensely monitoring higher risk women and avoiding unnecessary intervention in low-risk women.

Pregnancy is characterised by widespread metabolic changes [21,22,23]. Metabolomics, the quantification of molecules arising from metabolic processes, could improve the prediction of common pregnancy-related disorders [24]. We have recently shown that a targeted nuclear magnetic resonance (NMR)-derived metabolomics panel of 156 (mostly lipid) traits can improve the prediction of GDM, LGA and hypertensive disorders of pregnancy (HDP) in Born in Bradford (BiB), a large general population pregnancy cohort. This work was externally validated in the UK Pregnancies Better Eating and Activity trial (UPBEAT), a cohort of obese pregnant women [25]. We have also identified novel metabolite predictors for specific pregnancy outcomes using a mass spectrometry (MS)-derived metabolites platform in the Pregnancy Outcome Prediction study (POPs), which were externally validated in the BiB cohorts. Specifically, we found that the amino acid 4-hydroxyglutamate improves the prediction of PE in term pregnancy. A ratio of the product of the plasmalogen 1-(1-enyl-stearoyl)-2-oleoyl-GPC (P-18:0/18:1) and the steroid 5α-androstan-3α,17α-diol disulfate to the product of the carbohydrate 1,5-anhydroglucitol and the polyamine N1,N12-diacetylspermine was a better predictor of fetal growth restriction/SGA than the clinically validated biomarker soluble fms-like tyrosine kinase 1 and placental growth factor ratio (sFlt:PlGF) [26,27]. In the previous studies, we used serial samples from POPs, and the focus was identifying a small number of independently predictive metabolites that might form the basis of a targeted assay. However, these ratios do not form part of the analysis of this paper. In this paper, we use the entire MS metabolomic profile and an elastic net approach, with models developed in one cohort and validated in another.

To date, most studies have focused on a narrow range of metabolites and/or on a single pregnancy outcome, lack external validation and suffer from overfitting. The aim of this study is to determine whether metabolites included in an extensive MS metabolomics platform can improve the prediction of six common pregnancy outcomes (GDM, PE, GHT, SGA, LGA and sPTB) over established risk factors alone. Here, we validate the results externally and show metabolomics enhances prediction over maternal characteristics risk factors alone. The multimorbidity that exists between these disorders may mean a prediction tool for more than one could be clinically valuable. We therefore compare the overlap between traits across the prediction models for each outcome and whether a single prediction model can be developed to predict the occurrence of any of the six outcomes.

## 2. Results

The BiB 1000 and BiB 2000 sub-samples have similar distributions of age, BMI, parity and smoking, and, by design, both are around half White British and half Pakistani (Table 1). In contrast, around 95% of the POPs women studied are of White ethnicity. The BiB 1000 and BiB 2000 have higher smoking prevalence than POPs. BiB is a cohort of both multiparous and nulliparous women, whereas POPs is all nulliparous women.

Table 2 shows the number of predictors retained in each model during model training in BiB 2000. Of the total 723 predictors included in the combined risk factor (*n* = 5) and metabolites (*n* = 718) model, most were retained in the sPTB model and least in the GDM model. At least four of the five established risk factors were retained in the risk factor models for all outcomes except SGA. A full list of the retained predictors is available in Appendix A.

We found little overlap between the predictors retained in the combined risk factor and metabolite models across outcomes. There was an overlap of 41 predictors in the combined models for the GDM and LGA models, including 4-hydroxyglutamate, which was also retained for PE. There were five predictors retained in both the combined GDM and PE models, including 4′hydroxyglutamate and glycerol. There were 33 predictors retained for both LGA and SGA, including lanthionine, pipecolate, BMI, ethnicity and parity.

In both BiB and POPs cohorts, discrimination for GDM, PE, LGA and SGA improved with the addition of metabolites to the risk factor only model (combined model). The combined model AUC’s were similar for both cohorts, with good discrimination for GDM and LGA (GDM-(AUC (95% CI)) BiB 0.76 (0.71, 0.81) and POPs 0.76 (0.72, 0.81); LGA-BiB 0.86 (0.80, 0.91) and POPs (0.76 (0.60, 0.92)). Discrimination was improved for the combined models compared to the risk factors for PE and SGA, but the AUC was modest (PE-BiB 0.68 (0.58, 0.78) and POPs 0.66 (0.60, 0.71); SGA-BiB 0.68 (0.63, 0.74)) and POPs (0.64 (0.59, 0.69)). In BiB, risk factors alone were the best predictor of GHT: 0.74 (0.68, 0.80) compared to 0.72 (0.66, 0.79) for the combined model. The GHT models could not be validated in POPs due to an inadequate number of cases with metabolite data. Discrimination for sPTB was very poor for all models and in both cohorts, with a negligible improvement in the BiB 1000 testing set with the addition of metabolites to the risk factors model (0.54 (0.40, 0.67) increasing to 0.56 (0.45, 0.67)) (Figure 1). In POPs, the AUC was 0.50 for both models (Appendix A).

Calibration slopes showed good adherence of the predicted probabilities to the observed outcomes for the combined metabolite and risk factor models for GDM and SGA. Calibration was moderate for LGA. There was overestimation for GDM and underestimation for SGA and LGA as the intercepts show below (Figure 2, Figure 3 and Figure 4). Calibration was attenuated for the models in POPs (Appendix A). Calibration for the other outcomes was poor (Appendix A).

### Additional Analyses

In an additional analysis of all PTB in BiB (141/1441 cases in BiB 2000 training and 38/915 cases in BiB 1000 testing), results did not differ substantially from those for just sPTB (87/1441 cases in BiB 2000 training and 21/915 cases in BiB 1000 testing), with poor discrimination (Appendix A).

There were 828 women with ‘any’ pregnancy-related disorder in the BiB 2000 (training) and 301 with ‘any’ pregnancy-related disorder in the BiB 1000 (testing). The combined risk factors and metabolite model had very moderate discrimination, with an AUC for predicting ‘any’ pregnancy-related of 0.62 (0.58, 0.66). This is only a slight improvement on the discrimination for the clinical risk factor model 0.60 (0.56, 0.64). It did have good model calibration (Appendix A). The retained predictors include those retained in the models for GDM, PE and LGA, including glutamate, 4-hydroxyglutamate, glycerol and the nucleotide 5,6-dihydrouracil (Appendix A).

## 3. Discussion

In this study, we showed improved prediction for GDM and LGA using a model that combines mass spectrometry-assessed metabolites and risk factor predictors compared to established predictors only. We also found improved prediction of PE and SGA in the testing and external validation cohorts—but for these outcomes, discrimination was modest. These models were tested and externally validated in two independent cohorts. The BiB cohort is largely socioeconomically deprived (68% living in an area of the highest quintile of socioeconomic deprivation in England), includes women of different ethnicities and is ~40% nulliparous. POPs is more affluent (0% living in an area of the highest quintile of socioeconomic deprivation in England)(26), the majority are White women (~95% in this study) and all women are nulliparous. The validation of the models trained and tested in BiB in POPs suggest the models are not influenced by overfitting and are generalisable across diverse populations.

Calibration was good for the combined risk factor and metabolite model for GDM and SGA, and it was moderate for LGA. Calibration was consistently attenuated in the POPs validation. This would be expected, as the key determinant of calibration is the prevalence of the outcome in the underlying population, and re-calibration (by modifying the model intercept based on population prevalence whilst keeping all other parameters the same) is often used when comparing models between populations. We would therefore expect the calibration to be better in BiB 1000 than POPs, as it is from the same underlying population as BiB 2000. The marked differences between POPs and BiB in socioeconomic background, ethnicity and parity would contribute to making outcome prevalence very different. The risk factor model validation of LGA and SGA performed particularly poorly in POPs. This may be explained by participant characteristic differences between the two cohorts, such as ethnicity. Ethnicity is a good predictor of these outcomes in BiB. In the observational analyses in POPs, there is no association between ethnicity and LGA and SGA, possibly due to very little variation in ethnicity in POPs (with 95% of women being White).

Due to the known overlap in pregnancy-related disorders, we trained (BiB 2000) and tested (BiB 1000) the prediction models for ‘any’ pregnancy-related disorder. The discrimination for the combined prediction model was modest and not much improved from the risk factors model. This suggests that a single metabolomic prediction tool for ‘any’ pregnancy complication is unlikely to be feasible, despite the fact that calibration was good.

In our previous work, we found that NMR-derived metabolomics improves upon risk factors for the prediction of GDM, LGA, SGA and combined PE and GHT (hypertensive disorders of pregnancy—HDP). We reported the best discrimination for GDM and LGA, and our findings here suggest that metabolites from different platforms are valuable for the prediction of GDM and LGA [25]. In previous work in POPs and BiB 1000, we found that the amino acid 4-hydroxyglutamate was a novel predictor of PE, and the metabolite ratio described above was a better predictor of fetal growth restriction/SGA than a biomarker ratio used in the diagnosis of PE (sFlt1:PlGF) [27,28]. The focus of these previous studies was to identify a small number of independently predictive metabolites to be used in a targeted assay. The aim of this study was to evaluate predictive associations with a metabolomic profile using an optimised model development approach, elastic net. Therefore, the ratios were not used in this study.

The consistency in observations across the two distinct cohorts implies that some of these metabolites might be causal in determining the pregnancy complications studied. Consistent with our earlier published work, 4-hydroxyglutamate was one of the metabolites maintained in the combined model for PE in the new BiB 2000 cohort. 4′hydroxyglutamate was also retained in the combined models for GDM, LGA and in the ‘any’ pregnancy-related disorder combined model. The lipid glycerol was a retained predictor for GDM, PE and the ‘any’ pregnancy-related disorder combined model. These represent potential markers for future study. Despite these individual metabolites being maintained in prediction models for more than one outcome, overall, there was relatively little overlap between the retained models where we had good prediction for different pregnancy-related disorders.

These findings suggest that metabolites (from different platforms) contribute to the improved prediction of GDM, LGA, SGA and PE compared to the models using only established maternal risk factors. Whilst some studies have also shown value in metabolomics for predicting PTB, most of those have small sample sizes, have not compared predictive ability to established risk factors or not externally validated findings [29,30,31]. Our work, with external validation, suggests that PTB is not accurately predicted by established risk factors, metabolomic profile or the two combined.

The key strengths of this study are the internal and external (independent) validation and the exploration of a very large number of metabolite measures. We assessed the discrimination and calibration of our prediction models and found using metabolomics improved performance compared to established risk factors for GDM, PE, LGA and SGA. MS has the advantage of giving greater coverage of the metabolome than other methods such as NMR; however, it is more expensive (~150 GBP a sample as opposed to ~25 GBP a sample). Due to this, it is hard to find any other cohorts currently with MS data generated in pregnancy samples to strengthen our findings. None of these datasets represented random samples; therefore, future work should continue to validate these models in more general populations.

Despite trying to harmonise data between BiB and POPs, there were some differences in outcomes between the two. POPs performed metabolomics on a subset of cases of PE with more severe PE and excluded women with non-severe superimposed term PE from the case definition in the case-cohort design. There were only 12 cases of LGA in the POPs case-cohort, as LGA was not one of the outcomes of interest in the original case-cohort analysis. Therefore, the estimates have wide confidence intervals. In BiB 2000, we included any PTB as cases, whereas in POPs, the case definition included sPTB only. Therefore, we used sPTB in the main analysis, which included both BiB and POPs. However, it is reassuring that the results are consistent between the two studies. Whilst PE is the more severe form of HDP, GHT affects more women and is also associated with adverse perinatal and longer term outcomes. Independent external validation of the GHT model is important but could not be performed in POPs since GHT was not included in the case-cohort design in this population.

Ideally, we would have a prediction tool that could be used as early as possible in pregnancy and be repeated throughout for an updated risk assessment. This would allow women’s antenatal care to be tailored to their risk from early pregnancy. This was not possible in this study, as we have only one sample taken at 26–28 weeks’ gestation in BiB. Furthermore, all three of the studies used here were selected, non-random samples (Figure 5A–C). The models that we found to improve the prediction of pregnancy-related disorders need to be further tested on blood samples measured early in pregnancy and random samples of general populations of pregnant women. An interesting avenue for exploration would be to explore the relationship between metabolomic markers and the time of onset for the pregnancy-related disorders in this paper.

## 4. Materials and Methods

### 4.1. Participants

BiB is a population-based prospective birth cohort that recruited 12,453 women who had 13,776 pregnancies. Full details of the study methodology were reported previously [32]. In brief, most women were recruited at their oral glucose tolerance test (OGTT) at approximately 26–28 weeks’ gestation, which is offered to all women booked for delivery at Bradford Royal Infirmary, except those with known diabetes. Eligible women had an expected delivery between March 2007 and December 2010. Bradford, in the North of England, is one of the most deprived cities in the UK. In BiB, most of the obstetric population consists of women of White British or Pakistani origin (together accounting for 81%). Ethical approval for the study was granted by the Bradford National Health Service Research Ethics Committee (ref 06/Q1202/48).

**Figure 5 metabolites-11-00530-f005:**
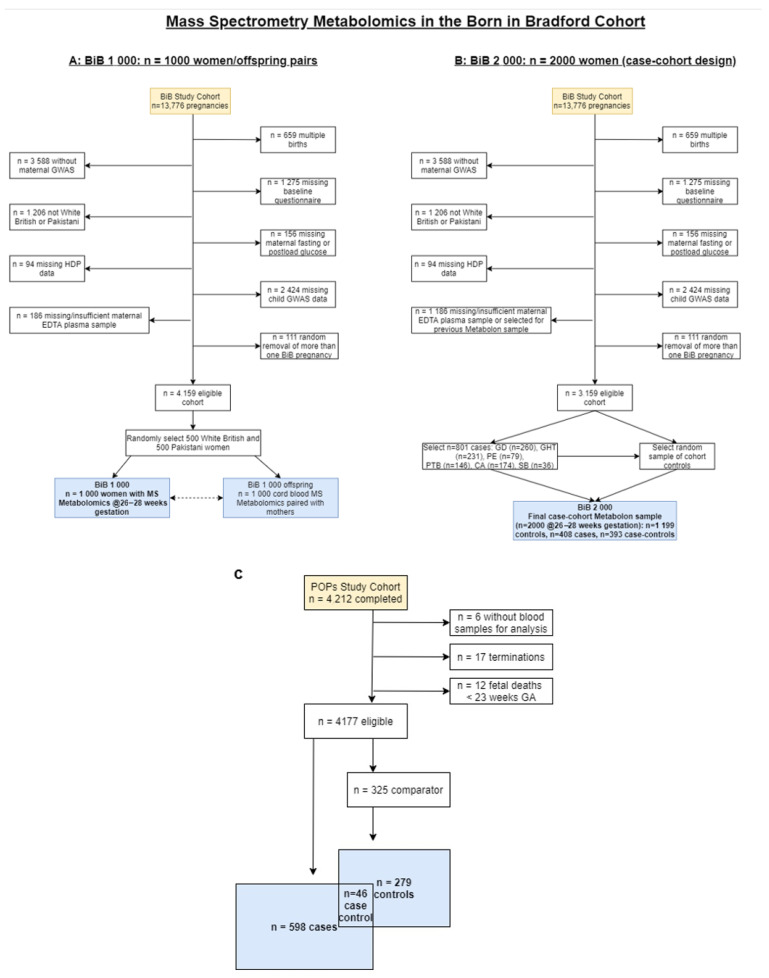
Born in Bradford flowchart: the selection of participants for mass spectrometry metabolomic profiling in the Born in Bradford 1000 (**A**) and 2000 (**B**). Abbreviations: MS, mass spectrometry; BiB, Born in Bradford; GWAS, genome-wide association study; EDTA, ethylenediaminetetraacetic acid; HDP, hypertensive disorders of pregnancy; GD, gestational diabetes; GHT, gestational hypertension; PE, pre-eclampsia, PTB, preterm birth; CA, congenital anomaly; SB, still birth. (**C**) Illustrating the flow of participants into the Metabolon datasets ((**A**) BiB 1000, (**B**) BiB 2000 and (**C**) POPs (*n* = 923) cohorts). Abbreviations: MS, mass spectrometry; BiB, Born in Bradford; GWAS, genome-wide association study; EDTA, ethylenediaminetetraacetic acid; HDP, hypertensive disorders of pregnancy; GDM, gestational diabetes; GHT, gestational hypertension; PE, pre-eclampsia, PTB, preterm birth; sPTB, spontaneous preterm birth; CA, congenital anomaly; SB, still birth; FGR, fetal growth restriction; GA, gestational age. (**A**,**B**) were taken from our data note by Taylor et al. [33] with permission.

The BiB metabolomics data have been described in detail previously in a data note [33]. In brief, 3000 women were selected for plasma MS metabolomic profiling using samples taken at 26–28 weeks’ gestation [32]. These 3000 women had profiling in two separate sub-samples of 1000 women and 2000 women. There was no participant overlap. Dataset 1 was completed in December 2017 and consisted of 1000 mother (pregnancy) and offspring (cord blood) pairs. These were randomly sampled from pairs where both had a suitable sample for analyses and belonged to one of the two main ethnic groups in BiB (Pakistani or White British) (Figure 5A,B and Appendix A). Only the maternal pregnancy metabolites are used in this study. Dataset 1 is referred to as “BiB 1000”. Dataset 2 (“BiB 2000”) was completed in December 2018 and consisted of 2000 mothers (Figure 1B and Appendix A) selected using a case-cohort design, including cases of GDM, PE, GHT, SGA, LGA, PTB, stillbirths and congenital anomalies, together with a randomly selected sub-cohort of the whole eligible cohort [28]. BiB 2000 was used for training the prediction models and the BiB 1000 for testing them.

External validation was undertaken in POPs, a prospective cohort study of unselected nulliparous women [19]. Those eligible were recruited at the Rosie Hospital, Cambridge, U.K., between January 2008 and July 2012. Cambridge is an affluent city in eastern England, and participants in POPs are nulliparous and mostly of White European origin. Ethical approval was obtained from the Cambridgeshire Research Ethics Committee (reference number 07/H0308/163). All study participants gave written informed consent. POPs also utilises a case-cohort design, previously described elsewhere (Appendix A) [19,28]. Only singleton pregnancies were included in this study.

### 4.2. Metabolomic Predictors

The untargeted MS analysis of BiB and POPs samples was performed at Metabolon, Inc. (Durham, NC, USA) using a modification of a previously described ultra-performance liquid chromatography (UPLC)–mass spectrometry (MS) method (UPLC-MS/MS) [27,28]. The platform consisted of four independent UPLC-MS/MS methods. This method provides relative quantification of >1000 metabolites in key classes: amino acids, carbohydrates, lipids, nucleotides, carbon metabolism, vitamins, xenobiotics and novel unidentified metabolites. Metabolite concentrations were quantified using the area under the curve of primary MS ions and were expressed as the means of the medians (MoM) value for all batches processed on the given day. Metabolon data are provided in a quantified (scaled) data set. Samples from the BiB 1000, BiB 2000 and POPs were run in different batches and are all analysed separately in this study (BiB 2000 for training, BiB 1000 for testing and POPs for external validation). In BiB, MS analyses were undertaken on ethylenediaminetetraacetic acid (EDTA) plasma taken around 26–28 weeks’ gestation. In POPs, MS analyses were undertaken on maternal serum samples from 12, 20, 28 and 36 weeks of gestation. In this study, we used the 28-week gestation timepoint for our external validation, as this best matched the gestation of the BiB samples. Previous work has shown that reproducibility in both serum and plasma is good. If the same sample procedures are used, either matrix should yield similar results [34]. Further information on the metabolomic data in both cohorts has been published previously [28,33,35]. The participant selection workflows are available in Appendix A. We included 718 metabolites in our models, which is the number of metabolites available in BiB 1000, BiB 2000 and POPs. A full list of the metabolites and their pathways included in this study as predictors is provided in Appendix A.

### 4.3. Risk Factor Predictors

We compared metabolomic prediction models to models of risk factor predictors that are routinely collected in antenatal care: maternal age, early pregnancy BMI, parity, ethnicity and smoking during pregnancy. This information was collected during recruitment or extracted from clinical records. In BiB, data on parity and weight were extracted from the first antenatal clinic records (around 12 weeks of gestation). Weight (kg, Seca 2 in1 scales, Harlow Healthcare Ltd., London, UK) and height (cm) were measured using established protocols at recruitment and used to calculate BMI. Parity was dichotomised as having experienced one or more previous pregnancy ≥24 weeks gestation or no previous pregnancy. Data on age, ethnicity and smoking were obtained from questionnaires administered by the BiB research fieldworkers at recruitment. These fieldworkers included people fluent in the common languages used by women undergoing antenatal care in Bradford at the time of recruitment, including Urdu and Mirpuri. In BiB, ethnicity was self-reported or obtained from primary care medical records and categorised as either White British or Pakistani (the only two ethnic groups who had metabolomic profiling and who together include over 85% of the BiB mothers).

In POPs, only nulliparous women were recruited [36,37,38]. Weight measured at recruitment (~12 weeks) was used for the calculation of BMI, and maternal age was defined as age at recruitment. Maternal height was measured at ~20 weeks. Maternal ethnicity and smoking were self-reported using a questionnaire at ~20 weeks in POPs, and ethnicity was categorised as White or non-White for the present analysis. In both studies, smoking was dichotomised as any smoking during pregnancy.

### 4.4. Outcomes

We examined predictive discrimination for six pregnancy-related disorders: GDM, GHT, PE, SGA, LGA and sPTB. In BiB, blood pressure and proteinuria measurements taken at any time during pregnancy were extracted from medical records. In BiB, GHT was defined as a new onset of elevated blood pressure (systolic blood pressure ≥140 mmHg or greater and/or diastolic blood pressure ≥90 mmHg or greater) after 20 weeks’ gestation on two or more occasions. In POPs, cases of GHT were not identified for inclusion in the case-cohort sub-sample on whom metabolites were assayed. Consequently, we were unable to explore the external validation of our GHT prediction models. In BiB, PE was defined as GHT plus clinically significant proteinuria, defined as 1 or greater ‘+’ on the reagent strip reading (equivalent to 30 mg/mmol) or greater on the spot urine protein/creatinine ratio (4). In POPs, PE was defined according to the 2013 American College of Obstetricians and Gynaecologists (ACOG) guidelines [28,39]. The selection of cases did not include non-severe superimposed PE associated with delivery at term, meaning PE in POPs likely reflects a somewhat more severe group than in BiB. In BiB, GDM was defined according to modified World Health Organization (WHO) definition used in clinical practice at the time: fasting glucose ≥ 6.1 mmol/L or 2 h post-load glucose ≥ 7.8 mmol/L [5]. In POPs, between 2008 and 2010, GDM diagnosis was based on diagnostic criteria adapted from the WHO recommendations: fasting, 1 h and 2 h glucose levels of ≥ 6.1 mmol/L, ≥ 10.0 mmol/L or ≥ 7.8 mmol/L, respectively. From 2011 onwards, these were replaced locally with diagnostic criteria adapted from the International Association of Diabetes and Pregnancy Study Groups’ recommendations: fasting, 1 h and 2 h glucose levels of ≥ 5.3 mmol/L, ≥ 10.0 mmol/L or ≥ 8.5 mmol/L, respectively [40].

In both studies, we used the Hadlock formula to derive gestational age and sex standardised birthweight percentiles as previously described [41,42]. SGA was defined as <5th percentile birth weight (being a more accurate measure of fetal growth restriction than the conventional 10th percentile) and LGA as ≥90th percentile birthweight. In both cohorts, sPTB was defined as the spontaneous onset of labour (i.e., without medical or surgical induction or elective Caesarean section) before 37 completed weeks of gestation. In POPs, there was additional criteria that delivery occurred after 24 completed weeks of gestation. Whilst this specific criterion was not applied to BiB, no births before 24 weeks gestation were included in the metabolomic profiling.

### 4.5. Statistical Analysis

All of the statistical analysis was performed in R 3.5.1, R.1.3.1 or STATA 15.1. In POPs, prior to this paper’s analysis, one woman’s BMI was imputed by sample mean and 15 women’s ethnicity was imputed by the most common category (White), since the proportion of missing values in these variables was very small. Aside from these imputations, all data were complete, and any participants with missing data on risk factors or outcomes were excluded from the analysis.

### 4.6. Comparison Groups in the Case-Cohort BiB 2000 and POPs

BiB 2000 and POPs sampled participants for metabolomic analyses using a case-cohort design, which increases statistical power (by including all cases for each outcome). This allows the analysis of associations of metabolites with multiple outcomes, with the comparison group for each cohort being the equivalent of a random sample of the underlying cohort [42]. In a case-cohort design, we only use the non-cases of the given outcome from the random sub-cohort as controls. Therefore, we remove oversampled ‘cases’ that are not a case of interest for this study.

### 4.7. Main Analyses

Only metabolites present in all three datasets (BiB 1000, 2000 and POPs) were included in the prediction models (*n* = 718). Prior to analysis, scaled and imputed (using the mean of the medians) metabolite values were log transformed and then converted to standard deviation (SD) units by subtracting the sample mean from their logged value and dividing by the sample SD of the logged values, within the prediction model. This results in some metabolites with very little variance. We applied a variance threshold in BiB of more than 440 unique values. This reduced the number of metabolites in the dataset from 1371 to 1074 in the BiB 1000 and 1241 to 1152 in the BiB 2000. We then only included metabolites present in all three datasets (BiB 1000, 2000 and POPs) and included these in the prediction models (*n* = 718). In the training dataset (BiB 2000), we optimised the elastic net penalized regression fit by tuning model hyperparameters, alpha and lambda, using 10-fold cross validation implemented with the *caret* package in R [43]. The hyperparameter tuning tests out different values of alpha and lambda and chooses a combination of both that results in the best performing model. Optimal hyperparameter value combinations were selected that maximised the cross-validated AUCs in the training set. The lambda parameter constrains the sum of the individual predictor coefficient values, performing feature selection. Feature selection is the process where the most relevant predictors, or features, are chosen for inclusion in a prediction model. Thus, non-informative predictors are removed from the final prediction model. Out-of-sample prediction performance was further assessed by AUC and calibration slopes in both the BIB 1000 test set and external validation in POPs. In all, we assessed the ability to discriminate pregnancy-related disorders by comparing AUC values for three models: (1) the established risk factors model, (2) the MS-derived metabolites model and (3) the combined established risk factors and metabolites model.

### 4.8. Additional Analysis

When studying preterm birth, POPs focused on those cases that were due to spontaneous preterm labour (sPTB). For concordance, sPTB was also used in BiB despite the availability of cases of non-spontaneous preterm birth. However, in an additional analysis, we also trained and tested a model for predicting any PTB, defined as any delivery < 37 completed weeks in the BiB cohorts irrespective of the onset of labour.

We wanted to see whether we could create an accurate prediction model for ‘any’ pregnancy-related disorder. We created a binary variable of ‘any’ of GDM, GHT, PE, LGA, SGA or sPTB cases compared with women who had experienced none of those. As in the main analysis, for these additional analyses, we trained the models in BiB 2000 and tested them in BiB 1000, assessed discrimination using AUC and calibration using calibration slopes.

## 5. Conclusions

To conclude, our results suggest metabolomics combined with established risk factors can improve the prediction of GDM and LGA. We validated our findings in an independent cohort. However, we acknowledge the need to validate these findings in a large independent sample of unselected pregnant women and examine their accuracy when measured earlier on in pregnancy. These findings show promise for the use of blood-derived metabolomics to improve the prediction of common pregnancy complications. Further validation would support the development of the tool using only the specific metabolites (rather than incurring the cost of the full panel) and testing its effectiveness in practice. Thus, our findings provide a promising evidence base for further research with the aim of being able to tailor antenatal care for women at risk of GDM and LGA.

## Figures and Tables

**Figure 1 metabolites-11-00530-f001:**
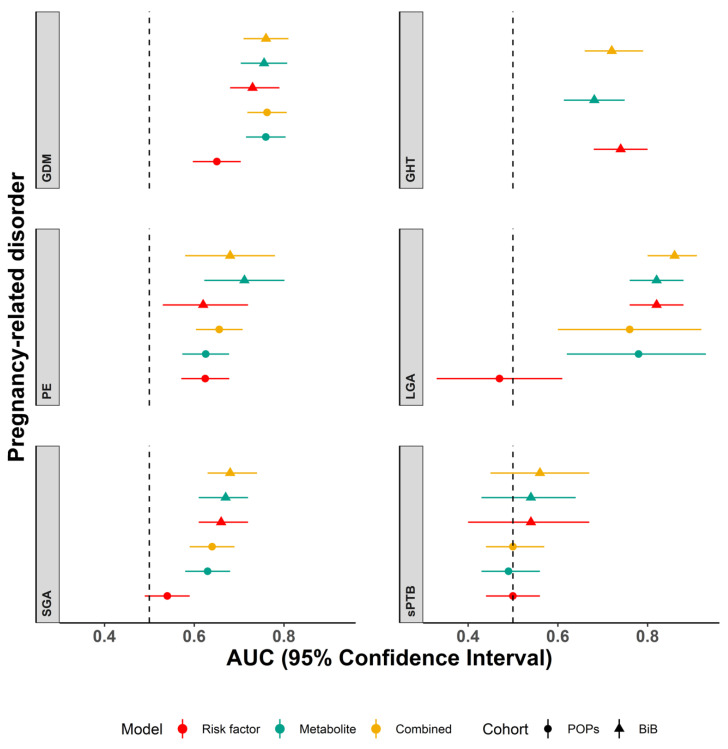
Predictive discrimination of models for each outcome. AUC and 95% confidence intervals are shown for established risk factor prediction models (red), metabolite models (green) and combined risk factor and metabolite models (yellow) trained in the Born in Bradford 2000, tested in the Born in Bradford 1000 (triangles) and externally validated in the Pregnancy Outcome Prediction study (circles). POPs did not have sufficient data on gestational hypertension for validation. Abbreviations: BiB, Born in Bradford; POPs, Pregnancy Outcome Prediction study; GDM, gestational diabetes; GHT, gestational hypertension; PE, pre-eclampsia; SGA, small for gestational age; LGA, large for gestational age; sPTB, spontaneous preterm birth.

**Figure 2 metabolites-11-00530-f002:**
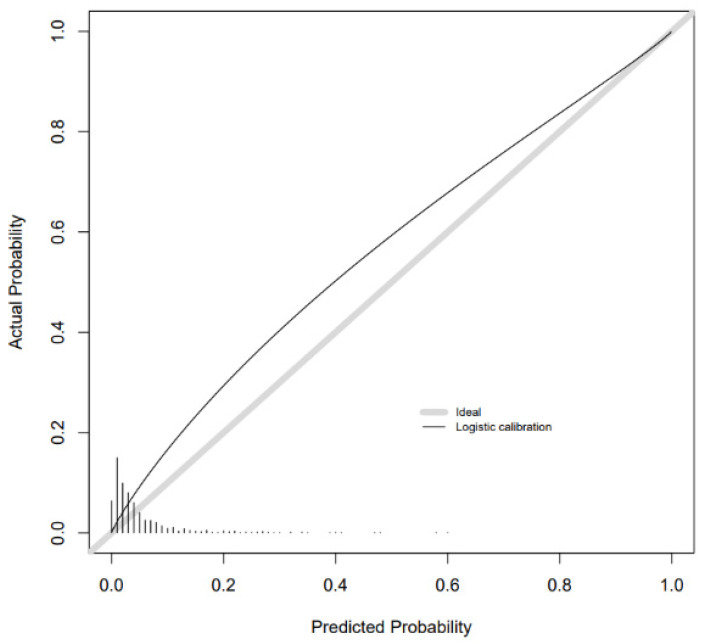
Calibration slope for GDM combined model in BiB 1000 testing sample.

**Figure 3 metabolites-11-00530-f003:**
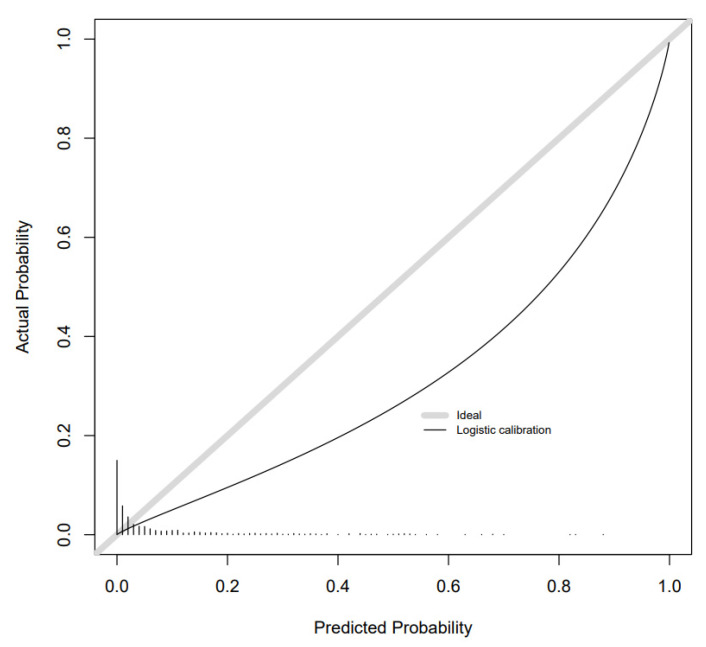
Calibration slope for LGA combined model in BiB 1000 testing sample.

**Figure 4 metabolites-11-00530-f004:**
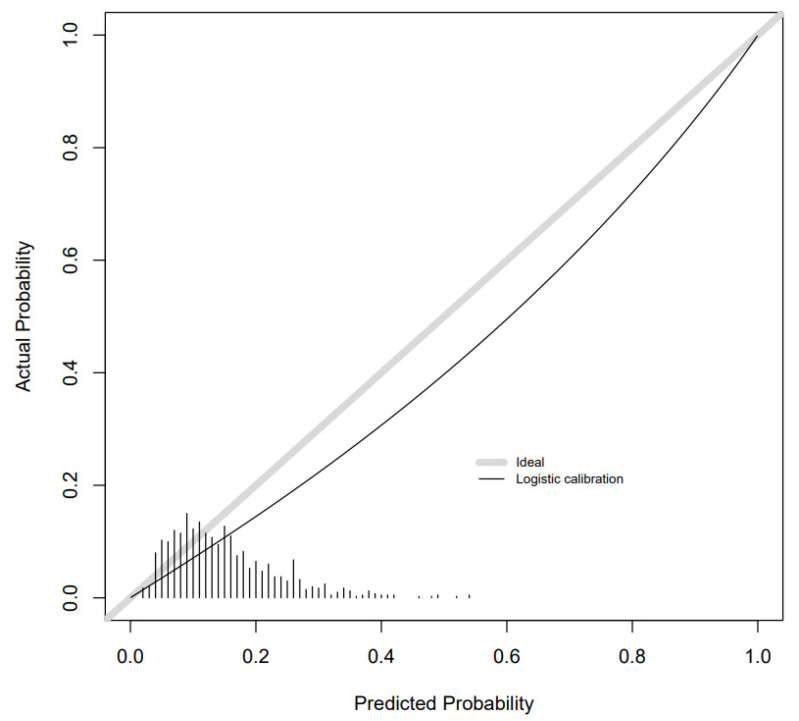
Calibration slope for SGA combined model in BiB 1000 testing sample.

**Table 1 metabolites-11-00530-t001:** Participant characteristics from the three participating cohorts: Born in Bradford 2000, Born in Bradford 1000 and the Pregnancy Outcome Prediction study.

Characteristic	Born in Bradford 2000	Born in Bradford 1000 *n* = 915	Pregnancy Outcome Prediction Study *n* = 827
Gestational diabetes (case/comparator) or *n* (%)	245/1350	84 (9.2)	172/295
Gestational hypertension (case/comparator) or *n* (%)	217/1375	64 (7.0)	6/300
Pre-eclampsia (case/comparator) or n (%)	74/1494	24 (2.6)	175/286
Large for gestational age (case/comparator) or *n* (%)	76/1425	37 (4.0)	12/294
Small for gestational age (case/comparator) or *n* (%)	260/1275	102 (11.1)	188/279
Spontaneous preterm birth (case/comparator) or *n* (%)	87/1441	21 (2.3)	98/297
BMI kg/m^2^ (mean (SD))	26.8 (5.8)	26.7 (6.0)	26.0 (5.3)
Age (mean years)	27.5 (5.6)	27.4 (5.7)	30.3 (5.3)
Pregnancy smoking, *n* (%)	378 (18.9)	159 (17.4)	119 (14.4)
Multiparous, *n* (%)	1213 (60.7)	581 (63.5)	0 (0)
White ethnicity, *n* (%)	933 (46.7)	456 (49.8)	787 (95.2)

Data in this table are complete. BiB 2000 and POPs used a case cohort design, i.e., they were over-sampled for cases. In these two studies, the total numbers vary depending on the outcome. For the distributions of risk factor predictors in this table, we used the overall mass spectrometry sample cohorts, *n* = 2000 for BiB 2000 and *n* = 827 for POPs (Appendix A). Because of substantial oversampling of cases in these studies, we do not give a prevalence (%) for the outcomes but rather give the numbers of cases and number in the comparison group for each outcome. The number of women in the comparator group varies per outcome, as some from the comparator group are always relabelled as cases. POPs did not have an adequate number of women with GHT; hence, no validation analysis was performed. Abbreviations: BMI, body mass index.

**Table 2 metabolites-11-00530-t002:** Number of predictors retained in each model developed and tested in BiB 2000 from total possible (*n* (%)). Percentages are rounded to the nearest whole number.

Outcome	Model (Retained Predictors/Total Number of Predictors Possible (%))
Gestational diabetes	Risk factor (4/5 (80%))
	Metabolite (81/718 (11%))
	Combined (82/723 (11%))
Gestational hypertension	Risk factor (4/5 (80%))
	Metabolite (28/718 (4%))
	Combined (75/723 (10%))
Pre-eclampsia	Risk factor (4/5 (80%))
	Metabolite (154/718 (21%))
	Combined (28/723 (4%))
Small for gestational age	Risk factor (5/5 (100%))
	Metabolite (66/718 (9%))
	Combined (65/723 (8%))
Large for gestational age	Risk factor (5/5 (100%))
	Metabolite (490/718 (68%))
	Combined (360/723 (50%))
Spontaneous preterm birth	Risk factor (4/5 (80%))
	Metabolite (587/718 (83%))
	Combined (328/723 (45%))

## Data Availability

Data are available upon request from https://borninbradford.nhs.uk/research/how-to-access-data/ accessed on 16 June 2021). The POPs study data are available from G.C.S.S. (gcss2@cam.ac.uk) upon reasonable request. Data requests will require a formal Data Transfer Agreement. Data are not publicly available due to the terms of the ethical approval.

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
