# Peer review of "Do Mass Spectrometry-Derived Metabolomics Improve the Prediction of Pregnancy-Related Disorders? Findings from a UK Birth Cohort with Independent Validation"

_metabolites, 2021, doi:10.3390/metabo11080530_

Round 1

Reviewer 1 Report

The article entitled “Do mass-spectrometry-derived metabolomics improve prediction of pregnancy-related disorders?” is an important finding. In this article, the authors are assessing risk factors based on clinical parameters of the cohorts and the expression of metabolites. It is an important finding, and the manuscript is well-written. Despite mentioning how metabolites are combined to analyze the risk factors, nothing much is mention about the expression levels of different metabolites in different groups.  Incorporating the expression levels of each metabolite in the groups and what made them select only a few metabolites are not well explained. Also, examining the hazard ratio after combining metabolites with the maternal risk factors will shed more light.

Reviewer 2 Report

The authors use metabolites information to pregnancy-related disorders. They demonstrated the ability of metabolomics to diagnose diseases with an elastic net penalized regression. In order to let readers understand the analysis process more clearly. I suggest the authors explain how to optimize the hyperparameters and impute the missing values. 

How to find the ratio of the product of the plasmalogen 1-(1-enyl-stearoyl)-2-oleoyl-GPC (P-18:0/18:1) and the steroid 5α-androstan-3α,17α-diol disulfate to the product of the carbohydrate 1,5-anhydroglucitol and the polyamine N1,N12-diacetylspermine was a better predictor is quite interesting. However, the authors did not mention how to identify the potential predictors/biomarkers.

Round 2

Reviewer 1 Report

N/A